# Status of Onchocerciasis Elimination in Gabon and Challenges: A Systematic Review

**DOI:** 10.3390/microorganisms11081946

**Published:** 2023-07-29

**Authors:** Elsa-Rush Eyang-Assengone, Patrice Makouloutou-Nzassi, Clark Mbou-Boutambe, Félicien Bangueboussa, Julienne Atsame, Larson Boundenga

**Affiliations:** 1Unité de Recherche en Ecologie de la Santé (URES), Centre Interdisciplinaire de Recherches Médicales de Franceville (CIRMF), Franceville BP 769, Gabon; patmak741@gmail.com (P.M.-N.); clarkmbou@gmail.com (C.M.-B.); bangueboussafelicien040@gmail.com (F.B.); 2Département de Biologie et Ecologie Animale, Institut de Recherche en Ecologie Tropicale (IRET/CENAREST), Libreville BP 13354, Gabon; 3Programme de Lutte Contre les Maladies Parasitaires, Ministère de la Santé du Gabon, Libreville BP 2434, Gabon; julienneatsame@yahoo.fr; 4Département d’anthropologie, Université de Durham, South Road, Durham DH1 3LE, UK

**Keywords:** Gabon, onchocerciasis, control, program, prevalence, vector, reservoir

## Abstract

Control and treatment programs (CDTI) have been set up nationally in all endemic countries to overcome the impact of onchocerciasis on the affected populations. However, Gabon must still succeed in setting up real onchocerciasis control programs. Here, various database articles have been used to provide the scientific community with a summary document showing the mapping of this disease in Gabon. The articles dealing with onchocerciasis, animal reservoirs, surveillance, and elimination were analyzed. Results showed that little research has been performed. Most studies are concentrated in one region (The area of Lastourville). In addition, we observed that the distribution of the disease varies significantly across the country. Indeed, specific environments present a hyper-endemicity of the disease, while others are meso and hypo-endemic. So, we found some departments with a prevalence ranging from 0% to over 20%; within them, villages had infection levels comprising 10% to 60%, indicating potential hotspots. Vectors activities were studied in some areas. This paper showed the challenges encountered in the country to eliminate this disease. One solution is a deeper understanding of the disease’s bioecology to establish effective health policies to eliminate onchocerciasis in Gabon effectively.

## 1. Introduction

Onchocerciasis is a parasitic disease caused by *Onchocerca volvulus* (*O. volvulus*). This filarial nematode is transmitted through the bites of black flies of the genus *Similium* that breed in rivers and streams [1]. Onchocerciasis is endemic in 26 African countries, where this infection alone accounts for almost 99% of the global disease burden [2,3]. The disease has been associated with a high impact on health and socioeconomic status due to blindness and stigmatizing scaly, itchy skin manifestations [4]. In 1974, the control of onchocerciasis in Africa began as a regional project known as the Onchocerciasis Control Program (WHO-OCP). The program was based on vector control with environmentally safe insecticides [5]. Although the WHO-OCP was successful, it was expensive and had technical challenges of insecticide resistance and black fly migration across the borders [6]. However, in 1987 the use of ivermectin became the mainstay for treating onchocerciasis [4].

Later on, the African Program for Onchocerciasis Control (APOC) was launched in 1995 to control onchocerciasis as a public health and socioeconomic problem [7]. Another objective of the program was to support establishing Community-Directed treatment with Ivermectin (CDTI) in all African areas where onchocerciasis was a public health problem [8]. After two years, CDTI was adopted as its core strategy and coverage [9]. This strategy was necessary because endemic communities were empowered to determine the timing of distribution, select their distributors, agree on the mode of distribution, and participate in the supervision of the Mass Drug Administration (MDA). In 2016, to maintain and build on the success of APOC, the WHO established a new structure named Expanded Special Project for Elimination of Neglected Tropical Diseases (ESPEN) to coordinate technical support for activities [10], sustaining the Neglected Tropical Diseases (NTD) control and elimination goals. ESPEN is constructed on three key pillars: accelerate pragmatic actions by providing data support to countries to use WHO evidence-based guidelines; intensify cross-cutting approaches through data, surveillance, and supply chain support for One-Health; and facilitate country ownership at national and subnational levels [11]. The elimination process starts with mapping onchocerciasis and instituting MDA with ivermectin, followed by monitoring and evaluation. In 2021, about 1,949,336 people lived in areas where treatment was stopped; Post-Treatment Surveillance (PTS) had been completed, and transmission was interrupted in 31 Implementation Units (IUs) in Colombia, Ecuador, Equatorial Guinea, Guatemala, Mexico, Sudan, Venezuela (the Bolivarian Republic of) and Uganda. Several countries stopped MDA in areas now under PTS [3]. Nevertheless, 23 countries reported treating 142.3 million people for onchocerciasis in the same year, representing 58.1% of global coverage [3].

Additionally, based on reports submitted to the WHO, six countries in the African Region (Central African Republic, Democratic Republic of the Congo (DRC), Equatorial Guinea, Gabon, Ghana, and South Sudan) canceled or postponed the 2020 MDA campaigns [2]. MDA is difficult to implement in Gabon due to the need for more information about the actual distribution of the disease in the country. Some studies carried out in the territory showed disparate levels of onchocerciasis transmission in some villages of Gabon [12]. Other studies showed the possibility of using different diagnostic tests to evaluate the endemicity of the diseases [13]. Nevertheless, these studies focused on the same region; those made throughout Gabon did not consider certain areas. Therefore, starting MDA in the country would only be possible if there is information on Gabon’s actual distribution of onchocerciasis. In addition, it would be necessary to have the prevalence of *O. volvulus* infection because Gabon is highly endemic for *Loa loa* (L. *loa*) infection [14]. It is known that special attention is required regarding the severe side effects that can occur after treatment with ivermectin in people with a high load of *L. loa* microfilaraemia, which may reduce the likelihood of establishing CDTI programs within the framework of the health system in Gabon [15]. Therefore, this review aims to have an update about onchocerciasis infection in Gabon to align with the WHO objectives of eradicating filariasis by 2030.

## 2. Materials and Methods

### 2.1. Search Strategy

Our study utilized four online databases: ProQuest, Web of Science, PubMed, and Google Scholar. To ensure accuracy, we conducted a literature review using a predefined protocol based on the preferred reporting items for systematic reviews and meta-analyses (PRISMA) [16]. The search process involved three keywords: “Onchocerciasis”, “AND”, and “Gabon”. To minimize bias in the number of articles obtained, the searches were carried out by six persons (Larson Boundenga (LB), Elsa-Rush Eyang Assengone (EREA), Patrice Makouloutou-Nzassi (PMN), Félicien Bangueboussa (FB), Clark Mbou-boutambe (CMB), Julienne Atsame (JA)) on the same day, using the same keywords. The search results for articles on onchocerciasis in Gabon were presented at a meeting of all researchers, and a comprehensive list of articles was created for data extraction.

### 2.2. Eligibility Criteria

Our goal was to produce a current map of onchocerciasis outbreaks in Gabon. This was achieved by analyzing the country’s research conducted on humans, animals, and insects. We included all studies conducted in various ecosystems in Gabon and considered scientific publications on neglected tropical diseases (NTDs) that mentioned onchocerciasis in their results. However, we excluded articles that were unavailable in full, those that referred to other articles, and studies unrelated to our objective.

### 2.3. Study Selection

All six authors evaluated the eligibility of each article’s title, abstract, and full text, and we extracted relevant background information. In cases where there were disagreements about whether to include or exclude an article, we held regular meetings to discuss the issue and resolve any disputes.

### 2.4. Data Extraction and Analysis

Once we reached a consensus through discussions regarding data extraction and interpretation, we selected articles that met our inclusion and exclusion criteria. The data from eligible studies included parasite species, vectors, villages, prevalence, study year, study area, habitat characteristics, and focus distribution. We categorized the results of our analysis based on the study’s objective and compared them with results from other authors who studied the same localities. Our data selection was based on the diversity of prevalence’s, the spatial distribution of outbreaks, and their classification as active outbreaks.

We carefully chose articles, abstracts, and reports in French and English to minimize bias. To further evaluate the potential for bias, we utilized the SYRCLE (Systematic Review Center for Laboratory Animal Experimentation) tool for animal studies, which consists of ten domains and six types of bias. This tool is highly effective in assessing the risk of bias.

Graphs were performed using R version 4.0.2 software [17].

## 3. Results

### 3.1. Available Data

After screening and analyzing, a total of 3821 articles were extracted from the four search engines ProQuest (1460), Web of Science (34), PubMed (17), and Google Scholar (2310) (Figure 1). After screening titles and abstracts and removing duplicates, we found 39 relevant articles on our topic. However, we included seven articles [12,13,18,19,20,21,22] that met all the criteria, while 32 were excluded for various reasons (for more information, see Figure 1). Of those, only one discussed the disease’s vector [20], while the rest were studies on onchocerciasis in humans. Unfortunately, we could not find any articles on the animal model (Figure 1).

### 3.2. Onchocerciasis Prevalence over Years

According to available data, 16,085 human blood samples were collected during studies on onchocerciasis in various regions of Gabon between 1992 and 2018 [12,22]. The prevalence of *O. volvulus* was highest in 1994 (42.27%) [12] and lowest in 2006 (7.7%) [20]. Over the years, fluctuations in the prevalence of the disease have been observed. There was an increase of the prevalence from 19.4% in 1992 [22] to 42.2 in 1994 [12]. There was a slight decrease of the prevalence’s from 1999 (33.8%) [19] to 2006 (7.7%) [20]. In 2007 the prevalence of onchocerciasis was 27% [13], and in 2012, the prevalence was 11.8% [21]. However, from 2012 to 2018, there was a drastic increase in infection rates by about 28.9% [18,21].

### 3.3. Areas at Risk of Onchocerca volvulus Infection

In 2015, onchocerciasis had a prevalence rate of 53.6% in Gabon, with some villages reaching up to 80% (as seen in Figure 2B) [18]. Through available data, we created distribution maps of *O. volvulus* infection, which helped identify the at-risk departments and the specific areas within them. Knowing the locations of villages provided greater accuracy on the department’s areas at risk [22]. The highest prevalence of infections was found in the villages of Dola and Mongo departments, followed by Tsamba-magotsi, Ogoulou, and Mulundu departments (Figure 2A,B). The Woleu department had a low prevalence rate of 0.14% (Figure 2A). However, some departments still require further study (Figure 2).

### 3.4. Data on Vectors and Reservoir

Based on available data, only one study has been conducted on the Simulium vector of onchocerciasis in Gabonese areas [20]. The study collected 30,397 blackflies of the *Simulium squamosum* species and estimated an average density of 506.6 bites per person per day. The highest densities were recorded between October and February during the rainy season, while the lowest occurred in August. Parturition rates varied from 17.8% in October to 41.0% in August when the Ogooué River was at low water [20]. The average rate for the year was 24.1%. Of the 4918 female flies dissected, 420 (8.5%) were found to harbor parasites at different stages, with a higher percentage during the dry season (June to August). The rate of infectious parous females over the year was 1.7%, with the highest and lowest values recorded in July and February, respectively [20]. These infectious blackflies carried 235 infective larvae stage (L3) in the head, or 47.8 L3 per 1000 flies [20]. The maximum number of L3 per 1000 flies was recorded in November, while the minimum was in February. The annual transmission potential was high at 2171 L3 per person. Maximum transmission intensity was noted in October and November, with PMT of 511 and 431 L3 per person, per month, respectively [20]. There is currently no study on the animal reservoir in the country that can be drawn.

## 4. Discussion

Onchocerciasis is one of the neglected tropical diseases for which chemoprevention is available to control morbidity and eliminate transmission. The WHO aims to eliminate the need for MDA of ivermectin against onchocerciasis in at least one focus in 34 countries by 2030. They also seek to achieve this goal for over 50% of the population in at least 16 countries and the entire endemic population in at least 12 countries. While challenges may arise, these elimination goals can be achieved by following the actions outlined in the NTD road map [3]. Since 1995, health policies have encouraged MDA by communities through the community-directed ivermectin treatment (CDTI) strategy, which is already well-established in several countries [23]. Thus, great progress has been made towards elimination of onchocerciasis as a public health problem in many endemic areas due to the efforts of Member States, implementing partners and major control and elimination programmes, such as the OEPA (1993–present), the OCP in West Africa (1974–2002), the APOC (1995–2015) and the ESPEN (2016–present) [3]. However, some countries, like Gabon, need help in setting up an effective strategy to fight against this disease. The absence or scarcity of data on onchocerciasis in Gabon, and the lack of any real support for the control program from the Gabonese government, are the reasons for the fragility or delay observed in the health policies implemented to fight onchocerciasis.

This literature review provides an overview of the available data on onchocerciasis in Gabon. The information in this document will help us to understand the challenges facing onchocerciasis control in the country. In the present review, the available data, though sparse, reveal that the few articles published on onchocerciasis in Gabon cover the period from 1992 to 2022.

In the present study, we observed that available data are scarce. Only a few published studies (seven articles) over 30 years provide usable data [12,13,18,19,20,21,22]. Moreover, on average, there is a two to five year elapse between two studies. However, the available data show a prevalence that varies over the years. Indeed, the results show a variation in the prevalence of infection over the years, with peaks observed in 1994 and 2018 at 42.2% and 40.7%, respectively [12,18]. These peaks in the population’s infection level by *O. volvulus* could be explained by the increase in investigation sites and the number of studies carried out by the research teams. Indeed, the prevalence of 19.4% observed by Chandenier et al. (1992) was the first description of the existence of foci of human onchocerciasis in Gabon in the regions of Makokou, Lebamba, Fougamou, and Lastourville area [22]. This first description opened the door to several other studies, enabling us to describe better the onchocerciasis situation in the country [12,13,18,20,21]. It has been observed that the prevalence of the disease increases as the number of areas studied increases. This could explain the rise in prevalence over the years [24,25]. Conducting studies in a larger area increases the chances of finding more sources of infection, this is why further research is crucial. However, it is worth considering that the variation in diagnostic tools used (such as parasitological examination, clinical examination, and serological test) and the target population can impact the accuracy of infection level estimation in different studies [12,26,27]. A study on onchocerciasis in the country was conducted through a parasitological examination involving an exsanguinated skin biopsy, a blood sample screening, and a clinical examination to search for nodules and ocular lesions [22]. Two years later, another study was carried out using a parasitological examination, a clinical examination, and a serological test (ELISA), which resulted in a 22.8% increase in prevalence rates [12]. The use of serological tests likely impacted the overall prevalence of infection in the study area. However, from 1994 to 2006, there was a progressive decrease in onchocerciasis prevalence, dropping from 42.22% in 1994 [12] to 33.8% in 1999 [19] and eventually reaching 7.7% in 2006 [20]. In 1999, a study supported by APOC/the WHO in Gabon used Rapid Epidemiological Mapping of onchocerciasis (REMO) in 140 selected villages [19]. However, 75 (53%) of the villages were inaccessible, reducing the target population’s size. Onchocercarial nodules were found in 22 of the 65 villages examined, and the nodule rate varied between 2.6% and 15.1% in all villages where nodules were detected. However, the decrease in infection rates (42.2% in 1994 to 7.7% in 2006) was not due to an effective control program but rather the inaccessibility of the selected villages for study, either by air or road [19]. This situation presents a genuine challenge for epidemiological studies that aim to map out onchocerciasis in the different ecosystems of Gabon, especially since we know that the Gabonese forest covers over 88% of the country [28]. Prevalence’s of onchocerciasis observed in 2006 and 2007 (7.7% and 27%, respectively) were obtained from a study conducted in Lastourville to evaluate the long-term impact of the activities of APOC. As for the two studies, these areas were not considered as being at risk of onchocerciasis because of the low rate of onchocercarial nodules [20,21]. Here we have observed that the prevalence of onchocerciasis from 1992 to 2018 does not vary linearly, but it varies depending on the study areas, their accessibility and the diagnostic tools used. So some studies were conducted in several areas of the country [18,21] and others in a single region [12,20] which are low-risk areas of onchocerciasis transmission. These results are therefore difficult to extrapolate to draw a map that faithfully represents the trend of the evolution of onchocerciasis over the years in Gabon. This constitutes one of the limits of our review, but it is a calling for further investigation in order to acquire a precise map.

Our findings have identified three distinct levels of *O. volvulus* infection in the country, categorized as hyper-, meso-, and hypo-endemic. Notably (Figure 2A,B), the departments of Ogooué-lolo, Nyanga, and Ngounié have the highest prevalence of nodules with some departments reaching a prevalence of ≥20% (Figure 2A). As such, we consider these areas to be hyperendemic and in need of targeted intervention. Meanwhile, most departments in the province of Ogooué-Lolo fall into the meso-endemic category, with nodules present in 10–20% of the population (Figure 2A). Therefore, we consider these areas to be hyperendemic and in need of targeted intervention. In addition, most of the departments in the province of Ogooué-Lolo fall into the meso-endemic category with nodule prevalences of between 10 and 20%. For these areas considered meso-endemic, if a mass treatment were to be carried out at the Gabon level, it would have been recommended to use ivermectin in these areas, too [29]. In addition, the majority of departments in Ogooué-Ivindo (Ivindo, Zadié, Mvoung, Lopé) and Haut-Ogooué (Mpassa, Lebombi-Leyou, Sébé-Brikolo) have nodule prevalence’s of between 0.1% and 10%, as do a few departments in Ngougnié (Louetsi-Wano and Boumi-Louetsi) and one each in Nyanga (Douigni) and Woleu-Ntem (Woleu) (Figure 2A). Those areas are hypoendemic for onchocerciasis, but they are nowadays also considered for MDA, as attention is now being paid in all hypo-endemic areas (nodule prevalence below 5%) to *O. volvulus* infection to prevent them from becoming future reservoirs that could weaken the strategies put in place in neighboring regions to fight against onchocerciasis [29,30]. APOC projects were initially confined to meso- and hyper-endemic areas but now extend to hypo-endemic areas [29]. However, we believe that when it comes to MDA or the establishment of CDTI in Gabon, it should be conducted cautiously due to the endemicity of loiasis throughout the country [14,18,31]. Furthermore, the prevalence of loasis varies according to the country’s ecosystems. Gabon has three major ecosystems: the forest, the savannah, and the lake region [14]. Thus, *L. loa* infection was higher in forest areas (Ogooué-Lolo, Ngougnié, Ogooué-Ivindo) than in savannah (Haut-Ogooué and Nyanga) and Lakeland (Ogooué-Maritime) [14]. Therefore, the risk of coendemicity between *L. loa* and *O. volvulus* infections is present in some regions of the country. It was observed that the prevalence of nodules varies depending on the type of environment. Therefore, within the same department, differences are observed in the prevalence of onchocerciasis in the villages of the department [32,33]. This is the case for the villages of Mongo (prevalence of nodules was ≥60%) and Basse-Banio (prevalence of nodules was 0%) departments in the province of Nyanga, which present different prevalence’s. The situation in Gabon raises questions about its role in fighting onchocerciasis in the sub-region. Without a clear policy and significant efforts, neighboring countries’ fight against this filariasis may be compromised.

Contrary to Gabon, the incidence of onchocerciasis in neighboring countries has favored the implementation of control strategies that have contributed to reducing or eliminating onchocerciasis [34,35]. Indeed, for many years CDTI has been implemented by countries hyperendemic to the infection [29,36,37,38]. For example, in Cameroon, where more than half of the area has been classified as at high risk of onchocerciasis [21,29], this strategy has had considerable effects in the health district of Thombell [34]. In this district, continuous treatment with ivermectin on the populations was followed for fifteen consecutive years. It resulted in a prevalence of less than 2% of people with microfilariae and 6.0% with onchocerciasis nodules [34]. The same observation was made on the island of Bioko in Equatorial Guinea (hyper-endemic for onchocerciasis) [35]. Indeed, before using ivermectin for mass treatment, the prevalence was 74.5%; eight years later, it had fallen to 38.4% [35]. In 2018, there was no evidence of current infection or recent transmission [35,39]. This strategy, which has proved effective, could be applied in Gabon as part of a national control strategy. However, it is interesting to note the emergence of resistance to ivermectin treatment, as observed in certain countries [40,41,42].

In addition to chemoprevention, vector control has been proposed as a complementary intervention strategy to eliminate transmission of onchocerciasis in endemic areas [43]. Understanding the dynamics of insect vectors in an endemic area is the prerequisite for developing effective strategies to break vector-human contact to implement effective monitoring of onchocerciasis control programs [44]. In Gabon, the analysis of available data reveals that no study has been conducted on *O. volvulus* reservoirs and that only one study focused on the entomological aspects of onchocerciasis [20]. This was conducted only in the Mahouya village in the Lastourville region located in the southeast of Gabon, an utterly forested area, thus revealing the presence of black flies *Simulium squamosum* [20]. The species identified in Mahouya village belongs to the black fly complex, *Simulium damnosum*, described as a species of rivers draining mountainous, forest, and savannah areas in West Africa [45]. Thus, after one year of capture in the Mahouya village, 30,397 black flies were collected. The number of Diptera reported in the single study carried out in Gabon was higher than those reported from Galabat (8202 flies) in Sudan [45] and Alabameta, Osum State in Nigeria (440 black flies) [46]. However, this was lower than Abu-Hamed’s reported number (39,394 flies) [45]. This variation in abundance rates can be attributed to various factors such as weather conditions (humidity, temperature, and precipitation), climatic changes (seasonal dryness and rain), physical and chemical properties of rivers, and human activities (deforestation and hydroelectric dams) [45,47]. Moreover, the maximum density of black flies was observed during the wet rather than the dry season. This is consistent with observations made in Nigeria [46], supported by the fact that increased rainfall will favor the breeding of black flies [48]. Thus, the result of the density showed that the daily bite rate was very high in this forest area of Mahouya (506.6 bites/man/day) compared to those observed in the Sudanian-Cameroonian savannah region (195 flies/person/day) and the rainforest (100 flies/person/day) [44]. The absence of effective strategies against vectors in the country could justify the daily biting observed in Mahouya. However, implementing a vector control strategy would reduce the pressure of vectors on populations living in endemic areas, reducing the transmission of the parasite, as has been observed in other countries. Indeed, according to Raimon and his co-authors (2021), the use of the “slash and clear” strategy has reduced the rate of *S. damnosum* bites by more than 90% in several regions that are endemic to onchocerciasis in the example of Maridi (South Sudan) [39,49,50].

Another control strategy would be the combination of MDA and vector control. In fact, according to Plaisier et al. (1997), combining these two methods would reduce antivectorial activity for about two years [51]. In Gabon, carrying out studies on vector bioecology in the different foci would be necessary because the lessons learned from other vector-borne diseases (VBD) demonstrate that a better understanding of vector behavior would provide valuable and essential information in the implementation of vector control methods [52,53]. The presence of living vectors in forest and savannah environments raises the question of an animal reservoir of *O. volvulus.* However, no studies have identified a reservoir other than humans. Thus, humans are considered the only definitive natural host of *O. volvulus*, although an adult worm of this species has been found in a gorilla [30]. On the other hand, a study carried out in the United States of America reports the presence of *O. lupi* in coyotes which are considered natural carriers that may contribute to the spread of this zoonotic nematode [54]. This observation could support the hypothesis of an animal reservoir of *O. volvulus*. Thus, additional studies on the search for potential reservoirs of this parasite should be carried out in Gabon to set up a “One Health” strategy (human-vector-reservoir) in the fight against onchocerciasis at the National level.

## 5. Conclusions

Onchocerciasis, or river blindness, is one of the diseases targeted for elimination by the WHO in the group of 20 diseases known as NTDs. Many endemic regions have made significant progress in eliminating onchocerciasis as a public health problem, and some have interrupted the transmission of the disease. This has been made possible by the implementation by these countries of effective programs to control and eliminate transmission. Gabon still has a long way to go in the fight against this disease. The country faces significant challenges, such as the unavailability of data on parasites, vectors, and animal reservoirs. Moreover, the quasi-endemicity of Loa-loa filariasis also complicates the fight against onchocerciasis. This suggests that further studies are needed in the Gabonese situation to complete and update data on onchocerciasis to understand the epidemiology of this disease in the country. However, improvement can be made through a concerted effort by the scientific community, effective control strategies, and the support of national health policies. Thus, all future strategies concerning the fight against onchocerciasis in Gabon settings must consider all aspects of the question through a One Health approach, focusing on consequences, people responses, and actions at the animal–human–ecosystems interface, implicating different research fields, including health personnel, parasitology, social sciences, ecology, wildlife, and medical entomology. Moreover, Gabon must adhere to the three strategic shifts set by the WHO: stronger accountability by shifting from process to impact indicators and accelerating programmatic action, a shift from vertical programming to intensified cross-cutting approaches, and a move away from partner-led to country-driven, country-owned work.

## Figures and Tables

**Figure 1 microorganisms-11-01946-f001:**
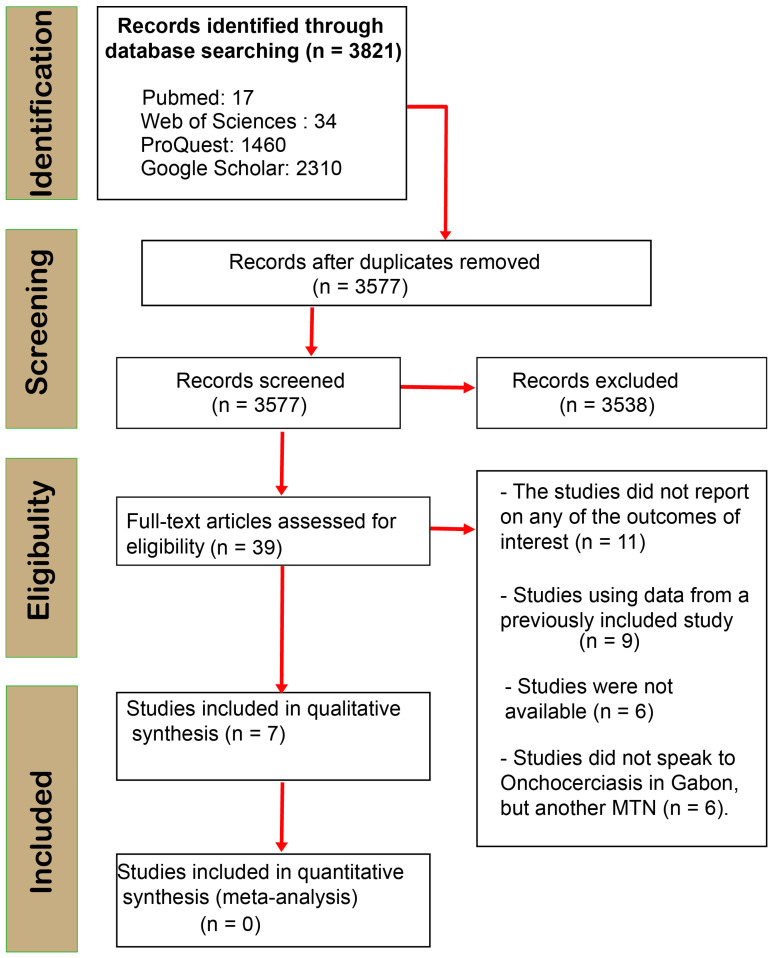
PRISMA flow diagram of search phases with numbers of studies included/excluded at each subsequent analysis stage.

**Figure 2 microorganisms-11-01946-f002:**
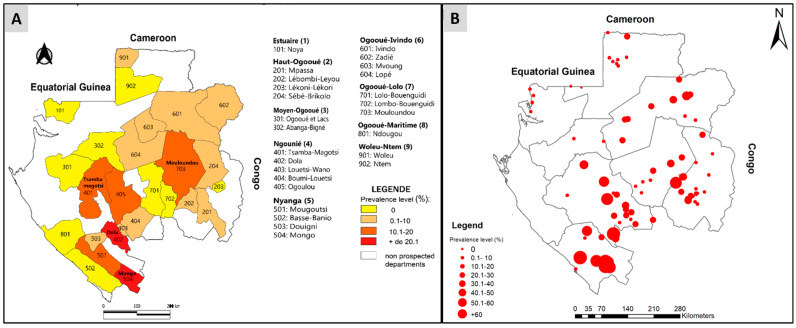
Distribution map of *Onchocerca volvulus* infection according to departments and villages of Gabon. (**A**) Distribution of onchocerciasis in Gabon according to department. (**B**) Distribution of onchocerciasis in Gabon according to villages.

## Data Availability

Please contact author for data requests.

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
