# Peer review of "Status of Onchocerciasis Elimination in Gabon and Challenges: A Systematic Review"

_microorganisms, 2023, doi:10.3390/microorganisms11081946_

Round 1
Reviewer 1 Report
All abbreviation should be explained upon first use, e.g.: Onchocerca volvulus (O. volvulus), only then the abbreviation can be used.
Six reviewers (LB, EREA, PMN, FB, CMB, and JA) evaluated the eligibility of each - It seems that the reviewers are coauthors of the manuscript. It would be more understandable to use either the full names of authors or explain the abbreviation. It would probably also be enough to just mention, that all authors participated on evaluating the eligibility.
According to available data, a total of 16085 human samples were collected during 138 studies on onchocerciasis in various regions of Gabon between 1992 and 2018. The prevalence of Wuchereria bancrofti was highest in 1994 (42.27%) and lowest in 2006 (7.7%). Over the years, there was a gradual decrease in infection prevalence from 1994 to 2006, followed by an increase to 27% in 2007 and then another decrease from 2007 to 2012 (11.8%). However, from 2012 to 2018, there was a drastic increase in infection rates by about 28.9%. These are the observed prevalence rates of O. volvulus infection in Gabon over the years (figure 2). – It is not clear from this text and Figure 2, if the numbers (or which ones of them) refer to Wuchereria or Onchocerca.
Author Response
-Toute abréviation doit être expliquée lors de la première utilisation, par exemple: Onchocerca volvulus (O. volvulus), alors seulement l’abréviation peut être utilisée.
Auteurs: Nous sommes d’accord et nous avons édité dans la nouvelle version manuscrite (par exemple voir ligne 30; lignes 91 à 93 pour nos modifications).
-Six évaluateurs (LB, EREA, PMN, FB, CMB et JA) ont évalué l’éligibilité de chacun - Il semble que les évaluateurs soient coauteurs du manuscrit. Il serait plus compréhensible d’utiliser les noms complets des auteurs ou d’expliquer l’abréviation. Il suffirait probablement aussi de mentionner que tous les auteurs ont participé à l’évaluation de l’admissibilité.
Auteurs: Merci, nous avons maintenant examiné vos suggestions dans la nouvelle version manuscrite (voir lignes 91-93).
-Selon les données disponibles, un total de 16085 échantillons humains ont été collectés au cours de 138 études sur l’onchocercose dans diverses régions du Gabon entre 1992 et 2018. La prévalence de Wuchereria bancrofti était la plus élevée en 1994 (42,27 %) et la plus faible en 2006 (7,7 %). Au fil des ans, la prévalence de l’infection a diminué graduellement de 1994 à 2006, puis a augmenté à 27 % en 2007, puis a diminué de 2007 à 2012 (11,8 %). Cependant, de 2012 à 2018, il y a eu une augmentation drastique des taux d’infection d’environ 28,9%. Ce sont les taux de prévalence observés de l’infection à O. volvulus au Gabon au fil des ans (figure 2). – Il n’est pas clair à partir de ce texte et de la figure 2, si les chiffres (ou lesquels d’entre eux) se réfèrent à Wuchereria ou Onchocerca.
Auteurs: Merci, nous sommes d’accord avec vous, maintenant nous avons édité cette section et le graphique dans la nouvelle version du manuscrit (voir lignes 145 à 156)

Reviewer 2 Report
In order to make the life of the readers easier, the reviewer suggests to report all the abbreviations in the “list of abbreviations” section.
Line 33. Could the authors be more specific? Could “world” be replaced with the continents or a specific area?
Report the disease as onchocerciasis instead of Onchocerciasis. The same for ivermectin.
Line 57. Maybe typos? Please revise.
Line 64. Carried on -> carried out
Revise all the references following the journal’s rules.
Line 71. After infection, add the appropriate reference. The same in line 74 after Gabon. The same for R software (line 118). The same in line 128 after “vector”. The same in line 138 after 2018. The same in line 264.
Line 99. Six reviewers -> all the authors
Line 113. Have all the articles been published in peer review journals?
Lines 123-125 need to be revised. The same for lines 205-207.
Line 126: remove “.”
Identification: 3577 article. Screening: 3577. Are you sure about that? So no duplicates in 4 different databases? This is really strange and must be clarified.
Line 138: what do the authors mean by “human samples”?
Section 3.2: a lot of data, 0 references. This is not possible. The same in sections 3.3 and 3.4. The same in lines 196-199.
Section 3.4: The meaning of L3 should be clarified.
Lines 201-202: 1992 is repeated twice.
Line 203. Use “,” instead of “;”.
Line 215: onchocerciasis
Line 222. The reviewer wonders what the authors mean by “We recommend mass treatment in these areas as well”.
Line 227: remove “.”
Line 238 = to another “one”
Line 266: replace “;” with “,”
The review is generally well-written, even if some sentences need to be revised. The reviewer suggests a deep check of English in order to make the review more fluent to read.
Author Response
Responses to Reviewer 2
-In order to make the life of the readers easier, the reviewer suggests to report all the abbreviations in the “list of abbreviations” section.
Authors : We are agree with you. A section named “list of abbreviations” has been created (see line 394-412).
-Line 33. Could the authors be more specific? Could “world” be replaced with the continents or a specific area?
Authors : Thank you for the suggestion. The authors added more specificity in the sentence (see line 32-33).
-Report the disease as onchocerciasis instead of Onchocerciasis. The same for ivermectin.
Authors: Thank you, the authors applied the changes in the whole text of the new version of the manuscript.
Line 57. Maybe typos? Please revise
Authors : It was a mistake ; It has been revised.
Line 64. Carried on -> carried out
Authors: Thank you, the authors made a revision on the new version of the manuscript.
Revise all the references following the journal’s rules.
Authors : Thankk you revisions have been made.
Line 71. After infection, add the appropriate reference. The same in line 74 after Gabon. The same for R software (line 118). The same in line 128 after “vector”. The same in line 138 after 2018. The same in line 264.
Auhors :We are agree with you, all the references have been added.
Line 99. Six reviewers -> all the authors
Authors : It has been changed
Line 113. Have all the articles been published in peer review journals?
Authors : Yes, all articles been published in peer journals. However, we also used some data from WHO reports on the disease.
Lines 123-125 need to be revised. The same for lines 205-207.
Authors: We Revised all those lines.
Line 126: remove “.”
Authors: “.” has been removed in line 126.
Identification: 3577 articles. Screening: 3577. Are you sure about that? So no duplicates in 4 different databases? This is really strange and must be clarified.
Authors: We agree with you, we realized we'd made a mistake on this part. Corrections have been made. There were duplicates because after examining the four databases we obtained a total of 3821 articles. However, after excluding the duplicates, 3577 items remained.
Line 138: what do the authors mean by “human samples”?
Authors : Authors meant « human blood samples, ». This has been corrected in the new version of the manuscript.
Section 3.2: a lot of data, 0 references. This is not possible. The same in sections 3.3 and 3.4. The same in lines 196-199.
Authors : We agree with the comment. We have added all the missing references in the text.
Section 3.4: The meaning of L3 should be clarified in the text
Authors: Thank you, it has been clarified (see line 192)
Lines 201-202: 1992 is repeated twice.
Authors : Thank you, one « 1992 » have been removed.
Line 203. Use “,” instead of “;”.
Authors: The suggestion has been took into consideration
Line 215: onchocerciasis
Auhors : It has been corrected
Line 222. The reviewer wonders what the authors mean by “We recommend mass treatment in these areas as well”.
Auhors: The idea has not been properly expressed in the text. The sentence has been reworded and explained in the new version of the manusript.
Line 227: remove “.”
Authors: Thank you, the dot has been removed.
Line 238 = to another “one”
Authors: Thank you the suggestion has been took into consideration
Line 266: replace “;” with “,”
Author : The suggestion has been took into consideration